# Repairing What Policy Is Missing Out on: A Constructive View on Prospects and Preconditions for Sustainable Biobased Economy Options to Mitigate and Adapt to Climate Change

André P. C. Faaij [1,2,3]

1   Energy & Sustainability Research Institute Gronigen, University of Groningen, Nijenborgh 6,
    9747 AG Groningen, The Netherlands; a.p.c.faaij@rug.nl
2   TNO, Energy Transition, Princetonlaan 6, 3584 CB Utrecht, The Netherlands
3   Copernicus Institute for Sustainable Development, Utrecht University, Princetonlaan 8a,
    3584 CB Utrecht, The Netherlands

**Abstract:** Biomass use for energy and materials is, on the one hand, one of the key mitigation options to reach the 1.5 °C GMT target set in the Paris Agreement, as highlighted by the IPCC and many other key analyses. On the other hand, particularly in parts of the EU, a strong negative connotation has emerged in public debate and EC policy, with a particular emphasis on the (presumed) displacement effect in markets and land use. This is a remarkable contrast because the reasons to use sustainable biomass, on the one hand, and the possibilities and synergies for supplying sustainable biomass, on the other, are underpinned with strong evidence, also providing insights on how displacement issues can be avoided. Sustainable biomass supplies can contribute 20–30% of the future global and European energy supply, leading to reduced overall mitigation costs, including realizing the net $CO_2$ removal from the atmosphere using BECCS concepts. This paper highlights which options, pathways and preconditions are key to achieving such a substantial contribution of sustainable biomass in future (2050) energy and material supply (with a focus on the European setting). By pinpointing how "biomass can be done right" and how important synergies can be achieved via better agricultural methods, the restoration of marginal and degraded lands and the adaptation of climate change, a different policy agenda emerges in sharp contrast to how a biobased economy has been framed in recent years. It is recommended that future policy priorities, particularly at the EU level, take a more integral view on the synergy between the role of biomass in the energy transition, climate adaptation and mitigation, better agriculture and the better use of land in general. Strategies to achieve such positive results typically require an alignment between renewable energy, and agricultural, environmental, mitigation and adaptation policies, which is a largely missing nexus in different policy arenas. Resolving this lack of alignment offers a major opportunity, globally, to contribute to the European Green deal and improve energy security.

**Keywords:** biobased economy; sustainability; synergies; mitigation; adaptation

## 1. Introduction

Biomass use for energy and materials is one of the most important mitigation options to reduce GHG emissions, as highlighted by various IPCC reports and scenarios over the years [1–4] as well as the IEA [5], IRENA [6], etc. Biobased options to replace fossil fuels include a wide range of existing and potential value chains and biomass resource—conversion technology combinations for delivering heat, electricity, gaseous and liquid fuels, biochemicals and biomaterials, and combinations of these. Biomass resources include organic waste, agricultural and forest residues, and cultivated biomass from various land categories, covering a wide range of (potential) annual and perennial crops. Furthermore, aquatic biomass (algae and seaweed) can add to the portfolio of resources.

The projected global contributions of biomass to the global energy supply in 2050 (and beyond) range between 100–400 EJ and about 10–30% of the total projected global future energy demand [1,7]; this contribution is comparable to today's biomass share (of which a considerable part is non-commercial biomass use, such as cooking fuel) to an extent that makes biomass a pillar of the future worlds' energy supply, similar to the role mineral oil has today. These ranges have been widely debated, analyzed and explained in the mentioned reports (see also: [8,9]). Furthermore, the mentioned references, backed by a very recent publication of the Energy Transition Commission of the UK, emphasized the importance of negative GHG emission options to meet the 1.5 °C target and biobased routes; they state that storing more carbon in vegetation and via the conversion of biomass and subsequent carbon capture and storage (BECCS) are the most important options to achieve this [10].

Given the state of affairs with the still-increasing global GHG emissions and the extremely fast emission reductions that are needed to keep the 1.5 °C GMT change target of the Paris Agreement in sight—in addition to the EC having policies in place to increase the use of sustainable biomass [11,12]—it is remarkable that, especially in the European Policy arena, biomass is largely negatively perceived; there is an especially fierce debate around the use of forest biomass and cultivated crops, which, in the end, negatively affects all relevant biomass resource categories and applications, including the important innovation processes to scale up and bring advanced technologies and biomass supply options to the market. Some of the main negative viewpoints dominating part of the public and policy debate in the European setting are, in a simplified form:

- Increasing the use of biomass waste streams and residues is, to a large extent, impossible because increasing their use will generally compete with already existing applications, thus leading to displacement effects. Specifically for forest biomass, the focus is on the (perceived) long carbon-payback times, and on the negative impacts on biodiversity, leading to pleas to look at forests only as a carbon storage option and not a source of biomass.
- The use of solar and wind energy is preferred over biomass use, making the role of biomass in the future energy supply marginal and unnecessary.
- The additional production of biomass via crops and using land will have two key negative effects: one is the displacement of current land use (either for food production or as nature areas) and the other is that intensified land use (e.g., to increase productivity) will lead to more agricultural emissions, water use and negative impacts on biodiversity. Furthermore, increased pressure on land can lead to increased food prices, land grabbing and the displacement of the current land owners (farmers).

The objective of this paper is to ask the counter question of whether these viewpoints are true and to what extent these (perceived) negative impacts of biobased options can be counteracted (with special attention given to the highlighted displacement risks). The paper achieves this via the assessment and synthesis of existing scientific research and reviews, targeting an integral overview of concepts and options that *can* lead to the large-scale and sustainable deployment of biomass for energy and materials, and contributing to (deep) GHG mitigation pathways. The review seeks a confrontation between these opportunities (and related conditions to realize them) and the recent and current policies and debate around biobased economy options. The main emphasis will be on the European setting, including the quantification of the potential role of biobased options for mitigating GHG emissions. The paper closes with concrete proposals to turn the current negative perceptions of biobased options in a different direction.

The paper covers the following main components, which are, in turn, the factors that determine to what extent biomass use for energy and materials leads to avoided GHG emissions: biomass resource availability, the future use of biomass for energy and materials, the GHG balances of biobased value chains, the net avoided GHG emissions of the value chains, and $CO_2$ capture and storage combined with biomass conversion and its impacts on carbon stocks.

Section 2 covers the main findings and viewpoints, to date, on the potentials of biomass use for energy and materials on a global and European scale, building on extensive assessment work in recent EU projects, the IPCC and other international bodies such as the IEA.

Section 3 covers analyses that focus on meeting the sustainability requirements for future (enhanced) biomass production, also with respect to the displacement risks (and avoidance of those), covering different biomass production systems, settings, regions and land-use categories.

Section 4 addresses the role of biomass in state-of-the-art GHG mitigation and energy scenarios on a global, European and (selected) country scale in conjunction with (all) other key mitigation options.

Section 5 provides a summary of the key preconditions for the sustainable production and use of biomass for the different main categories and settings, and policy recommendations that confront the current state of affairs around biobased economy options. The preconditions are compared to the required versus the existing policies and policy frameworks, resulting in the identification of gaps in and opportunities for better policies.

## 2. Biomass Supplies: Land-Use Scenarios and Potential for Biomass

The main categories of biomass have been distinguished, covering biomass waste, residues from agriculture and forestry, and cultivation on agricultural, pasture, marginal and degraded lands. The biomass resources (agricultural residues, forest biomass, cultivated crops and organic waste) that can be made available over time depend especially on the extent to which improvements in agricultural productivity allow for the use of productive land for biomass crops to be grown without displacing food. Furthermore, the use of marginal land and the mobilization of residues and forest management are key parameters. The future biomass resource supply, therefore, comes with sizeable ranges. More up-to-date estimates of biomass resource potentials take sustainability criteria with respect to land use, GHG balances, environmental and socio-economic impacts into account [13]. These criteria provide important preconditions for a future sustainable biobased economy.

### 2.1. Global Level

Biomass is seen as one of the key renewable energy options to displace fossil fuels and contribute to the mitigation of GHG emissions on a large scale [1,2,9,14]. Many scenarios that describe how a low-carbon future in 2050 can be achieved project substantial shares of biomass in the future global energy supply, ranging between 10% to over a third, adding a contribution of up to 300 EJ in many scenarios in the second half of this century [2] versus a projected total global energy use between 800 EJ and over 1500 EJ [15].

Global biomass energy potentials are found to range between 100 EJ and over 500 EJ/yr in 2050 (compared to a total global primary energy use of about 570 EJ today, with increasingly demanding preconditions to realize the higher values of this range, e.g., with respect to improvements in agricultural efficiency and sustainability standards [8,15]. It is Crucial in these figures is that future food demand, water limitations, biodiversity protection, and food demand are taken into consideration. Improvements in agricultural efficiency and crop choice are essential, too, especially perennial cropping systems that offer net lower input (nutrient) requirements, permanent soil cover and higher carbon stock build-up. In addition, genetic diversity and options for, e.g., trees and grasses are very large. Figure 1 (which is selected for its clarity and extensive global author team, although similar outcomes and ranges have also been reported in other recent assessments, such as in [16]) summarizes these insights on a global scale, based on a large number of studies and including estimates of the global potential availability of biomass residues from agriculture and forestry.

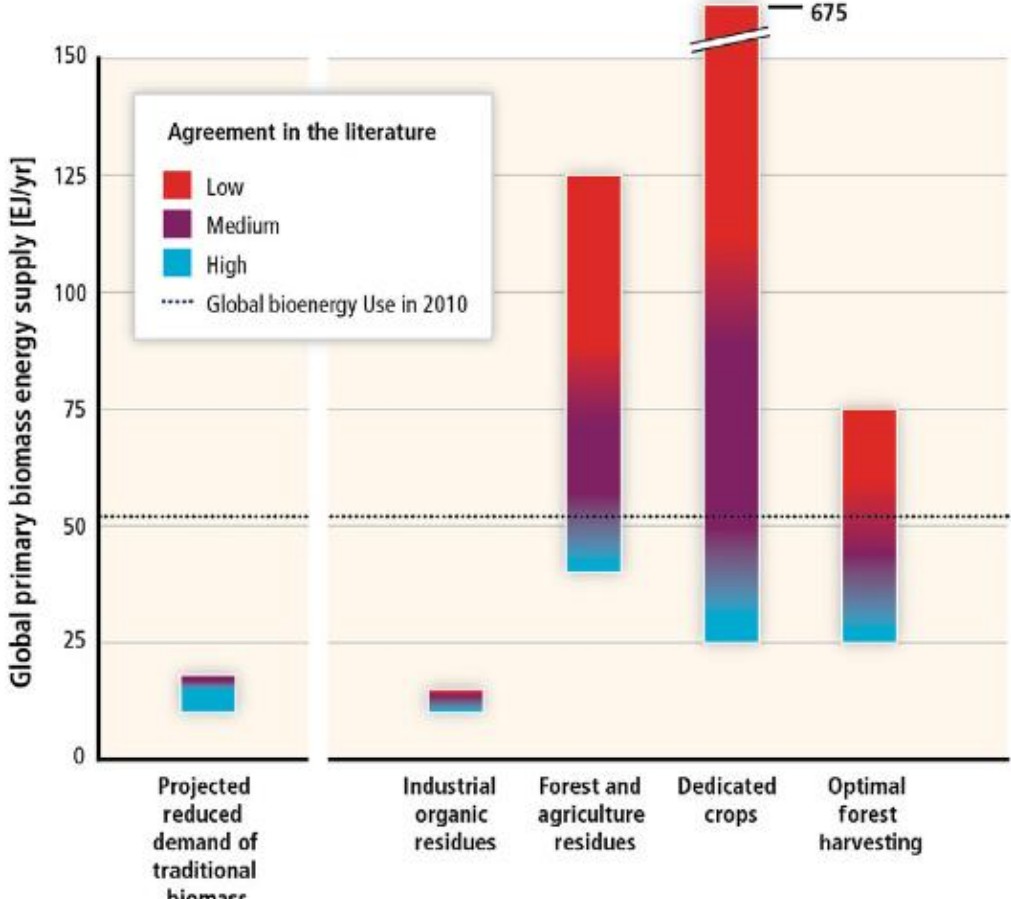

**Figure 1.** Global technical primary biomass potential for bioenergy by main resource category for the year 2050. The figure shows the ranges in the estimates by major resource category of the global technical primary-biomass potential for bioenergy. The color grading is intended to qualitatively show the degree of agreement in the estimates, from blue (all researchers agree that this level can be attained) to purple (medium agreement) to red (few researchers agree that this level can be attained) [8].

Recent work on global biomass resource potentials by Daioglou et al. [9,17] included all the major parameters on overall land-use change, carbon stock impacts, the net availability of residual biomass, etc. and analyzed different future scenarios concerning developments in land use and agricultural management (SSP1–SSP3; the different so-called socio-economic pathways, as defined by the IPCC), and concluded the following:

*"Biomass has an important role to play in future energy supply, irrespective of technological development or climate goals"*. Biomass forms at least 8%, and up to 35% of total primary energy supply by 2050 in all the baseline and mitigation scenarios presented, with its contribution increasing in mitigation scenarios. *In scenarios meeting ambitious "Paris style" climate targets, bioenergy makes up 26–35% of primary energy in 2050 and 32–50% in 2100, primarily used in the transport and power production sectors*. After 2050, bioenergy use is increasingly combined with CCS, providing so-called negative emissions, which are very important if the strict emission constraints of ambitious climate targets are to be met.

*High land requirements for energy production, together with other types of land use, run the risk of causing land-use change emissions per unit of energy above those of fossil fuels*. Ensuring an energy crop supply with low emission effects requires increases in the productivity of both energy and food crops, as well as livestock. These increases outpace the growth in food demand (including growing meat consumption). Optimizing the land required for food production (especially pasture) would allow for the availability of large volumes of

highly productive land for biomass production at low LUC emissions. Improved crop yields would also increase the supply of residues while lowering their costs.

The analysis highlights the favorable nature of SSP1 implementation where strict land-utilization constraints, improved crop yields and the reduction in extensive pastures may allow for low LUC/ILUC emissions of biomass production, as well as land-based mitigation; this is in contrast with the opposite SSP3, where substantial amounts of biomass are used, but with much-reduced GHG mitigation impacts due to indirect emissions and impacts on carbon stocks.

## 2.2. European Level

Scenario analyses on future European land use in relation to agricultural production and modernization are covered in recent assessments [18,19] as well as by the Joint Research Centre of the European Commission [20], in which the works of de Wit et al. [21,22] and Fischer et al. [23] provide one of the most integrated studies on agricultural production scenarios, their underlying variables and their implications for land-use change and carbon stocks.

Substantial biomass resource potentials in Europe can be developed, but again, at the same time, a considerable number of sustainability criteria need to be fulfilled to do so responsibly. The key risk to tackle is the displacement of land use (indirect land-use changes), but also of markets, because if biomass resources are taken away from existing markets, the supplies need to be produced elsewhere, as well. Improvements in the productivity of agriculture, livestock and forest management, as well as the use of marginal and degraded lands (not used for food production), can prevent this risk [14,21,22]. Table 1 shows the results from analyses on potential future land availability in Europe when existing yield and productivity gaps in food production systems are (partly) closed and efficiency improvements in livestock are realized; 7–52 Mha of arable land and 10–19 Mha of pasture lands could be released in Europe, while meeting future projected food demand. Then, such land surfaces may be used for growing additional crops and for the reforestation of managed grasslands. The 3.3–15.8 EJ range reported in Table 1 can be delivered when is the case. The wide range illustrates both the magnitude and the dependence on the extent to which and how fast yield improvements may be realized.

**Table 1.** Synthesis of European biomass potential. The time frame covers the period 2030–2050 (relevant for RED directive and for 2050 full decarbonization). The relevant geographical zone for the potential is EU-28, including analyses that cover the Ukraine and some other European countries that are not part of the EU-28.

| Biomass Category | Mha | EJ | Mtoe |
|---|---|---|---|
| Cultivated biomass | 7–52 Mha arable land 10–19 Mha pasture land | 3.3–15.8 | 79–377 |
| Agricultural residues | N.A. | 1.9–2.8 | 45–67 |
| Forest biomass | N.A. | 0.2–7.3 | 5–174 |
| Biomass waste streams | N.A. | 1.7–5 | 40–119 |
| **Totals** | | **6.2–30.7** | **148–734** |

1 EJ = 23.9 Mtoe (1000 Mtoe = 41.9 EJ, 1 Mtoe = 41.9 PJ, 1 toe = 41.9 GJ).

Defining forest biomass availability is not straightforward, as it includes logging residues, as well as the industrial by-products of wood processing. The latter includes residues from logging, bark and chips from sawmilling, and black liquor from the pulp industry. Three main categories are often distinguished: primary forest residues (such as thinnings, but also plantation wood), secondary forest residues (bark, sawdust, etc.) and tertiary residues including consumer waste and recycled building materials (see also the analyses of Mantau et al. [24] and Haninnen et al. [25]. A study by Smeets & Faaij [26] on

forest biomass resources highlighted this. The economic–ecological potential for Europe was estimated to amount to 405 $Mm^3$ under ecological criteria, which were more stringent than those considered in other potential categories.

Clearly, not one figure can be given for the forest biomass potential. The range obtained from this review lies between 0.2 EJ and 7.3 EJ (in the longer term). The low estimate includes criteria stating that basically all residue material is left in the forest. Most studies, however, agree that a supply of about 4 EJ could be mobilized by around 2030, and depending on the actions taken in forest management and the criteria applied, there is growth potential of up to 6 EJ or 7 EJ. This is confirmed, in turn, by the review of Ruiz et al. [19] who present figures on the combination of wood products, forest residue potential and wood processing residues in 2050 of between 1.3 EJ and over 11 EJ. It should be noted that wood products account for over 3 EJ in the maximum estimate. This can be compared to the reported wood use in 2013 of 485 $Mm^3$ (or about 4.2 EJ) in total, of which about 50% (about 2.1 EJ) was used for energy in the EU. The higher potentials in the longer term are especially explained by the expected higher demand for forest products in the underlying economic scenarios. De Wit et al. [21] also specifically focused on the possible cost–supply curves for forestry residues. As confirmed in [19], the bulk of the potential could be available at below 5 EUR/GJ.

The key conclusion from this extensive assessment of the literature and modeling of the results is that the potential biomass availability in Europe in 2050 lies between 7 and 25 EJ (which can be compared to a future projected energy use of about 70 EJ, [27,28]). A division in the different categories is made between cultivated biomass crops (depending on land availability), agricultural residues (well-analyzed), forest biomass (high variability depending on forest management) and organic waste streams. The latter category includes sludges, residues from various industries, manure, etc. Table 1 summarizes the compiled key data on future biomass resource potentials in Europe.

The overall estimation focus lies in the use of lignocellulosic biomass because lignocellulosic resources and (future) value chains deliver the highest net energy potentials and the best GHG balances overall [29,30]. This means that perennial crops (trees and grasses) are especially considered for the energy potential of crops. Biogas and related (wet) biomass resources (such as manure, swill, sludge and agrofood waste) are a separate category because of their high moisture content, which makes them more suitable for conversion technologies such as anaerobic digestion.

### 2.3. Factors That Are Given Limited Attention in Biomass Resource Potential Studies: Aquatic Biomass, Alternative Protein Supplies, Limiting Food Waste

Aquatic biomass (micro-algae and seaweed) is usually not included biomass resource potential analysis estimations. Nevertheless, seaweed is likely to make a significant contribution to biomass supplies towards 2050 [20]. However, the possible amounts and economic performance are, at this stage, still rather uncertain. The future contribution of these resources is still quite uncertain, although medium-term cost levels for seaweed production in the North Sea may become attractive [31].

A potentially big factor in reducing future land demand for food production is the growth of alternative protein supplies (e.g., produced via micro-algae, and shifts from animal protein to proteins derived from crops). If successful on a large scale, such supplies could lower the dominant land-use factor (pastures and arable land use for fodder crops to feed livestock) considerably, although such analyses are, so far, not linked to future biomass resource potentials. Lowering food waste in general due to poor logistics and storage, as well as end-use, represents more potential to lower net land demand for future food needs. The FAO has estimated, in the past, that up to one third of primary food production is lost during its production, distribution and final use, an indication that improvements in this respect could have sizeable moderating effects on future agricultural land use [32].

A key factor in future land use is climate change. Negative impacts on agricultural production worsen as GMT change increases, and this will also reduce the potential to

increase agricultural yields. On the other hand, the necessity for the adaptation to climate change impacts to avoid erosion and desertification and improve water-retention functions, a.o., can be achieved with more (permanent) vegetation cover, reforestation and better control of the production factors for agriculture [33]. Partially, this can also lead to increased biomass availability over time as a "side effect" of the necessary increased adaptation efforts. The net balance of all these mechanisms for future (sustainable) biomass availability is, however, understudied and deserves major attention in science and policy.

## 3. Sustainable Biomass Production Systems and Their Impacts

The following graphic (Figure 2), reworked from the SRREN report of the IPCC [1] shows how different aspects are linked together; biobased options are at the nexus of land use, and thus, agriculture, as are forest cover and other (e.g., marginal and degraded) lands and the global carbon balance (including the removal of $CO_2$ and the sequestration of carbon), with different impacts at different scales. The key notion brought forward by this graph is that biomass production can result in negative impacts and can be the result of poor strategies and policies, as well as positive impacts and synergies that can be achieved when various preconditions and sustainability requirements are met. This links to the available reviews of sustainability frameworks and certification systems and overviews of potential impacts of biobased value chains. Such frameworks and criteria for biomass production typically cover land use, environmental impacts (such as biodiversity, water, nutrients, soil quality, energy and GHG balances) and socio-economic impacts (economic performance, employment, rural economies, indirect effects such as potential displacement, etc.) [34,35]. Of growing importance is the increasing influence of climate change on land use and the urgency of adaptation measures for agriculture, forestry and land cover in general [4].

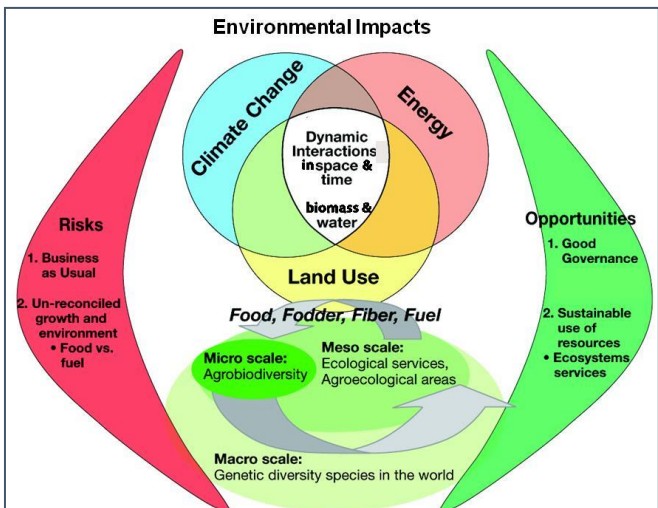

**Figure 2.** Nexus position of biobased economy options in climate, land use and energy, with impacts on macro-, meso- and micro-scale that depend on the way biobased value chains are implemented (derived from: [1]).

Refs. [8,34,36] present a rich overview of the variability of impacts that biomass production and use for energy (and materials) can have, depending on how this is implemented and managed. On basically any relevant sustainability criterion (whether these cover environmental or socio-economic impacts), the implications can be positive or negative, depending on crop choice, the management of land, the organization of the supply chain, etc. Furthermore, the notion that optimal solutions will differ from place to place is important. There is not one optimal biomass resource, and the combination of agricultural and land management with crop choices and cultivation methods determines the overall performance and the range of impacts. At the same time, this basic notion also makes clear

that in many cases, tailor-made solutions can be identified and implemented to produce biomass production with optimal environmental performance. Well-established sustainability frameworks and criteria, as developed under the RSB, GBEP and other key frameworks, give clear guidelines on how this can be achieved on local and regional levels, including by minimizing the displacement risks (see, e.g., [8,35]). The quantitative analyses of different cropping systems in relation to crop management, regional conditions and the land-use scenarios for the surrounding agricultural and livestock systems have become increasingly available for different countries and regions.

The FAO/UNEP/UNIDO framework for sustainable biofuels, the Global Assessments and Guidelines for Sustainable Liquid Biofuel Production [36], already presented an integrated approach for analyzing and quantifying the potential and environmental and socio-economic impacts of biomass for energy, taking specific national and regional conditions into account. Three countries, Mozambique, Argentina and the Ukraine, were analyzed using that framework ([36–38]) using spatio-temporal-explicit tools to make land-use scenarios for different agricultural development pathways (i.e., methods and levels to improve productivity). By fixing the land base for agriculture, and thus, fixing forest-cover and nature areas, any potential for future biomass production is limited to potential surplus lands. The environmental impact assessment shows that the impacts are related to the biophysical and socio-economic conditions in the region and the characteristics of the supply chain (mainly crop selection), and are dependent on the scenario conditions. Most negative environmental impacts occur when native vegetation is converted to bioenergy plantations. Generally, positive environmental impacts occur when abandoned cropland or degraded land is used for bioenergy plantations. The negative impacts can partly be reduced, or even turned into positive impacts, by taking adequate management measures (which include precision farming, mixed cropping stands such a agroforestry, cover crops, no-till farming, improved genetics, etc.). Some of the socio-economic impacts are directly related to the design and the management of the project, and can therefore not be assessed ex ante. Other socio-economic impacts such as the impact on social well-being and local prosperity are directly linked to the number of hectares and the total investment in the region. The authors of [39] conducted a similar and detailed analysis for Argentina, and in [38], a more refined analysis on the integral GHG balance of land use in the Ukraine is presented.

With the changes in agricultural and forest management towards more productive systems, the negative impacts with respect to the emissions of fertilizers, agrochemical use, biodiversity, and water and soil quality can be avoided to a large extent, or can result in co-benefits compared to current land use, particularly when monoculture production systems are replaced. Such preconditions and criteria have been well-covered by state-of-the-art sustainability frameworks and criteria [40,41] and the earlier reviews of [42], which outline integral systems with indicators, and thresholds for the full range of environmental and socio-economic sustainability criteria. In the paragraphs below, some detailed examples for specific regions are summarized.

### 3.1. Examples of Integrated Studies of Biomass Resource Potential including iLUC Prevention

State-of-the-art insights show that the ways in which productivity is increased—e.g., via "simple" increased input levels of fertilizers and the use of agrochemicals versus precision-farming techniques, which achieve far lower emissions (fewer greenhouse gas emissions, as well as reduced leaching and use of fertilizers and agrochemicals) with higher yields—have a major positive influence on overall environmental performance (see, e.g., [43,44] for a detailed analysis of different management schemes in Western Poland, and similar results for Rumania and Hungary [45,46]).

Gerssen Gondelach et al. [43,44] build on the analyses of future biomass production potential in Europe by de Wit et al. [21,22] by focusing on an agricultural region in Poland. Figure 3 presents the key results of a quantitative impact analysis on the total GHG balance of all land-use activities in that region when the productivity of current agriculture (and

livestock) is increased via different intensification methods (varying from simply increasing inputs such as fertilizers to best practices using precision farming, minimizing emissions), combined with the use of calculated surplus land for miscanthus production (in turn, assumed to be used for the production of second-generation ethanol). The results show not only that significant bioenergy potential without indirect land-use change is possible, but also that substantial GHG savings occur as a result of conventional agriculture due to better management practices. As can be expected, 'sustainable intensification' delivers the best overall result, with substantially lower GHG emissions per unit of agricultural output, as well as substantial increases in soil carbon.

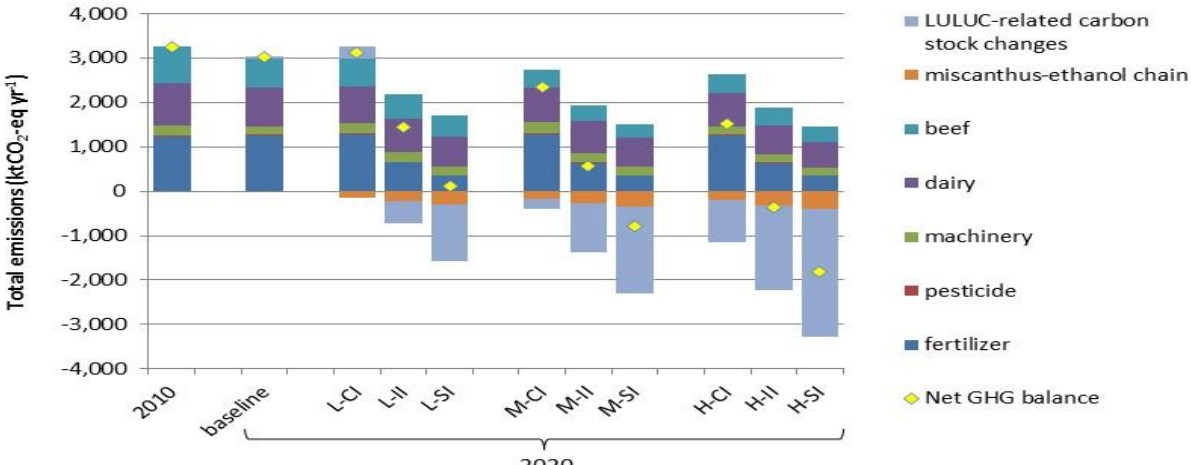

**Figure 3.** Total and net annual GHG emissions for 2010 and the baseline and ILUC mitigation scenarios in 2020. Emissions from the miscanthus–ethanol value chain. The equilibrium time for soil carbon stock changes is 20 years (taken from: Gerssen Gondelach et al. [43,44]). ILUC prevention scenarios: L—low; M—medium; H—high. Intensification pathways: CI—conventional intensification; II—intermediate sustainable intensification; SI—sustainable intensification.

Palm oil is seen as one of the most critical vegetal oil production systems because of its direct linkages to deforestation in, e.g., Indonesia and Malaysia. There is no doubt that the uncontrolled expansion of palm oil on cleared rainforest and peatland has a disastrous effect on GHG balances and sustainability (see also [47]). However, ref. [48] also concluded that the production of palm oil on marginal grasslands in Indonesia can deliver the opposite result; high avoidance of emissions, combined with additional carbon stock build up compared to the original land cover. In the Indonesian context, considerable amounts of such land are available that could accommodate the substantial expansion of palm oil production outside forest areas. Although slightly more expensive, the socio-economic impacts on these regions are likely to be (very) positive because of the minimal revenue that the marginal grasslands deliver. Van der Laan et al. [49] and Verstegen et al. [50] also highlighted that up to certain levels, the expansion of palm oil in Indonesia is possible without the loss of forest cover when combined with increasing agricultural and forest plantation productivity in Kalimantan, and the identification of areas where and pathways by which this may best be achieved.

Ramirez et al. [51] analyzed the possibilities for future palm oil expansion in the Colombian context under strict sustainability criteria. It was found that there is considerable improvement potential in crop- and value-chain management, which can lead to substantial GHG savings combined with lower production costs.

The combined improvement of agricultural management—in particular, livestock management and, again, the use of surplus productive land for bioenergy crops—was analyzed; this was combined with the quantification of the impacts, in total, for the entire region on GHG balances, economic performance, water use and biodiversity. Agricultural intensification is considered an important measure for making surplus agricultural land

available for energy-crop production, mitigating ILUC and improving the GHG emissions of biomass value chains. The intensification of cattle production has been identified as a crucial option in Colombia to free land for other purposes, such as biomass feedstocks for energy production. Table 2 (taken from [41]) gives an integral overview of the results obtained, comparing different levels of the intensification of current agricultural management and comparing oil palm, sugar cane and acacia trees. Overall, across the board, both environmental and socio-economic benefits can be achieved within the boundaries of the current land used for agriculture.

**Table 2.** Key impacts of agricultural intensification and bioenergy production in 2030 for 4 scenarios compared to 2018 for the Orinoquia region. Op—oil palm, Sc—sugarcane and AC—acacia [40]).

| Performance Indicators | Ref | Low | Medium | | | | High | | | |
|---|---|---|---|---|---|---|---|---|---|---|
| | Net Agri-Changes [a] | Net Agri-Changes [a] | Net Agri-Changes [a] | Net Bioenergy Changes [b] | | | Net Agri-Changes [a] | Net Bioenergy Changes [b] | | |
| | | | | Op | Sc | Ac | | Op | Sc | Ac |
| LUC [c] (change in natural vegetation) | $--$ | $-$ | $+$ | $+$ | $+$ | $+$ | $++$ | $++$ | $++$ | $++$ |
| GHG emissions [d] | $--$ | $-$ | $++$ | $++$ | $+$ | $+$ | $++$ | $++$ | $+$ | $+$ |
| Biodiversity [e] (change in species abundance) | $--$ | $-$ | $+$ | | $+/-$ | | $+$ | | $+/-$ | |
| Water use [f] | $--$ | $--$ | $+/-$ | $-$ | $-$ | $-$ | $+$ | $--$ | $--$ | $--$ |
| Net Present Value (revenue/ha) | $+$ | $+$ | $++$ | $++$ | $+$ | $+$ | $++$ | $++$ | $+$ | $+$ |

Signs: The signs indicate an increase ($+$) or decrease ($-$) in the impact compared to 2018 where $+$ is a positive change; $++$ is a very positive change; $-$ is a negative change; $--$ is a strong negative change; and $+/-$ is a negligible change. Abbreviations: Op—oil palm; Sc—sugarcane; Ac—acacia; [a] Agricultural changes refer to the effects caused by food crops and cattle production. [b] Bioenergy changes refer to the effects caused by energy-crop production on surplus land from agricultural intensification. It is assumed that all surplus land in the Orinoquia region is used either for oil palm, sugarcane or acacia, causing the same impact since there is no variation in the hectares used for energy cultivation. At the national level, oil palm crops are developed in the potential areas according to the land suitability map. [c] Land-use changes are analyzed considering the cover type and agricultural area under 2018 conditions for the Orinoquia region, considering this year as the current situation. The percentage of surplus land is the relationship between the total agricultural area currently in use in the Orinoquia region (6.8 Mha) and the surplus land obtained from the intensification of that agricultural land. [d] GHG emissions are evaluated based on the results of Sections 3 and 4. [e] For biodiversity in the medium and high scenarios in the Orinoquia region, agricultural intensification contributes to an increase in species abundance, mainly due to the reduction in the impact of increased cattle production and the conservation of biodiversity in natural vegetation, as assessed in Section 5. [f] Water use in agricultural production includes irrigation water for perennial food crops (i.e., plantain, cassava and oil palm for cooking oil) over the dry season. Moreover, it includes cattle water intake. In the medium and high scenario, water use for bioenergy considers irrigation during the dry season for the respective energy crops (i.e., oil palm, sugarcane and acacia).

Similar results are obtained for Brazil [37] and other countries and regions [33]. These also include examples of annual crops (often food crops), which are often qualified as less suitable for larger-scale biomass production due to the need for better-quality land; their production can also be realized by meeting the aforementioned sustainability criteria and increasing productivity, as demonstrated [45,46] for regions in Hungary and Romania.

Additionally, for Jathropha, despite being a crop with which a range of unfavorable results have been achieved, can be deployed with mostly benefits when implemented as a complimentary crop next to food crops and in a smallholder setting [52].

Although still less well-studied and monitored, mixed-cropping concepts, including agroforestry (with notable possibilities to increase biodiversity [53] and improve soil quality); rotation agriculture with, e.g., grass refining as one component, increasing overall soil productivity and delivering additional biomass yields; and winter cropping and intercropping (e.g., Camelina as a cover crop leading to additional vegetal oil production) are all examples of more productive, more environmentally benign and more resilient land management options that lead to additional biomass output, as well [33].

The IPCC has also highlighted the importance of such restoration schemes to contribute to removing net $CO_2$ from the atmosphere, restoring ecosystem services and delivering sustainable biomass for (advanced) applications. Ref. [4] includes an insight stating that a land surface the size of India is required for reforestation and restoration, to be used for biomass feedstock by 2050 to contribute the 1.5 °C pathway. This is also highlighted in the ETC report [10], which provides an overview of the negative GHG emission options on a global scale. Additionally, in Europe, such win–win schemes offer major opportunities, for water retention, soil restoration and, for example, to remove heavy-metal contamination from soils (e.g., phytoremediation; see, for example, [54]).

What is relatively understudied but of rapidly increasing importance is the desirability of increased and permanent land cover (such as via reforestation) to adapt to climate change. Dealing with weather extremes such as extreme rainfall makes water retention in landscapes increasingly important (as was concluded after the floods in summer of 2021 in Germany and Belgium), but counteracting desertification in the south of the continent is also of increasing relevance. A side-effect of such schemes, also to keep the forest vegetation where it is (which can be mixed stands of trees, grasses, etc.) is a sustained increase in biomass production. This is as much a top priority to address for the scientific research community as it is for market players and policy makers.

Agricultural Residues

Batidzirai et al. [55] present a key example of an integral assessment of the feasibility of sustainably mobilizing maize and wheat residues for large-scale bioenergy applications in South Africa by establishing sustainable residue removal rates and the cost of supply for different production regions. A key objective was to refine the methodology for estimating crop-residue harvesting potential for bioenergy use, while maintaining soil productivity and avoiding the displacement of competing residue uses. Under the current conditions, the sustainable bioenergy potential from maize and wheat residues in South Africa was estimated to be about 104 PJ. There is potential to increase the amount of crop residues to 238 PJ (in the improved scenario) through measures such as no-till cultivation and the adoption of better cropping systems. These estimates were based on total residue requirements ranging from 2–6 t/ha, depending on the soil type and crop management. About 96% of these residues are available at below 1.5 USD/GJ. In the improved scenario, up to 85% of the biomass is available at below 1.3 USD/GJ. Establishing sustainable crop-residue supply systems in South Africa could start by utilizing the existing agricultural system to secure a supply and a functional market. It would then be necessary to incentivize improvements across the value chain, such as shifting to no-till cultivation, improving agricultural management systems and crop yields, improving animal feed conversion efficiency, double cropping, employing contractors, and establishing pre-processing facilities and logistics infrastructure. This study provides a template for how sustainable agricultural-residue availability can be determined, organized and optimized over time.

Cruder but comparable insights have been obtained for Europe [21,22]. Research in, e.g., Denmark has included the monitoring of nitrogen and phosphate balances in relation to soil characteristics to determine sustainable removal rates [56]. Zhang et al. [57] and more detailed work on the Löss platea by Liu et al. [58] deployed the methods (with more refined data) of Batidzirai for China; they concluded that 226 Mt (3.9 EJ) of agricultural residues could be collected annually and maintain the current SOC level (MCSS scenario), compared to 116 Mt (2.0 EJ) for maintaining at least 1% (MSS scenario) and 24 Mt (0.4 EJ) for maintaining at least 2% (HSS scenario) of the SOC level at present. With increased crop yield and no-tillage management in 2050, the resource potential can be increased to 514 (8.9 EJ), 383 (6.6 EJ) and 117 Mt (2.0 EJ) under the IMCSS, IMSS and IHSS scenarios, respectively. No-till cultivation combined with improved crop yields could significantly reduce the amount of residue put into the soil and increase residue yields, leading to higher sustainable potential for these residues. The overall message is that agricultural residues,

with proper management, can deliver a substantial and increasing amount of biomass for biobased application in the coming decades.

Daioglou et al. [59] analyzed sustainable residue availability from agricultural production using both a (global) scenario analysis and IAMs; they projected a relatively stable total sustainable supply for the different SSP scenarios of around 100 EJ for agricultural and forestry residues combined.

### 3.2. Forest Management Biomass Resources and GHG Mitigation

Forest management plays a critical role in whether forests are effective carbon sinks and stable carbon pools. Both forest operations (e.g., replanting and thinning) as well as safeguarding the entire forest area (e.g., in a country) and carbon stocks are important in this respect. Creutzig et al. [8] highlight that for forest biomass, carbon-payback time is another concern; carbon accounting at the stand level, wherein the accounting starts when the biomass is harvested for bioenergy, naturally finds upfront carbon losses that are found to delay net GHG savings by up to several decades (carbon debt). Assessments over larger landscapes report both forest carbon gains and losses, which delay the GHG reduction benefit, as well as reductions in forest sink strength (foregone carbon sequestration), reducing or even outweighing, for some period of time, the GHG emission savings from displacing fossil fuels. In short, biomass that would otherwise be burned without energy recovery, rapidly decomposing residues and organic waste can produce close-to-immediate GHG savings when used for bioenergy; this is similar to increasing the biomass outtake from forests affected by high mortality rates. When slowly decomposing residues are used, and when changes in forest management to provide biomass for energy causes reductions in forest carbon stocks or carbon sink strength, the GHG mitigation benefits are delayed, sometimes by many decades. Conversely, when management changes in response to bioenergy demand so as to enhance the sink strength in the forest landscape, this improves the GHG mitigation benefit.

In large, managed forest estates, the management activities in one stand are coordinated with the activities elsewhere in the landscape, with the purpose of providing a steady flow of harvested wood. While the carbon stock decreases in stands that are harvested, the carbon stock increases in other stands, resulting in a landscape-level carbon stock that fluctuates around a trend line that can be increasing, decreasing, or remain roughly stable. Changes in the management of forests to provide biomass for energy can result in both losses and gains in forest carbon stocks, which are determined by the dynamics of the management operations and natural biotic and abiotic forces.

Therefore, sustainable forest biomass availability depends on many factors, but those factors are also well-understood and can (with a focus on production forests and plantations) be optimized for different and combined purposes. Duden et al. [60] and Jonker et al. [61] describe the dynamics of the 'fiber basket' in SE US and find that increased demand for biomass for, e.g., the wood pellet market is likely to lead to increased forest cover and productivity in SE US. Additionally, IEA Bioenergy highlights, based on extensive assessment work and an international review, that sustainable forest management can deliver excellent GHG balances and increased carbon storage, and can meet stringent sustainability criteria [5]. Carbon-payback time discussions are often focused on the plot level, assuming that whole trees are used for energy, but this perspective is in sharp contrast to real forestry management in the Northern Hemisphere, where forest cover and productivity have increased over the past few decades. The popular assumption that forests contribute more to mitigating climate change by storing carbon (permanently) is strongly opposed by the current real world statistics of increased forest fires and diseases propelled by climate change. Mature forests tend to take up less and less carbon over time (as has also been observed for the Amazon rainforest recently), while a productive forest can deliver continuous increases in carbon uptake combined with storing carbon below and above ground. When managing forests, keeping them resilient and productive is probably the best strategy to maintain (and increase) forest cover during this century.

### 3.3. Use of Marginal, Under-Utilized, Saline Lands, Degraded Lands and Contaminated Lands

The technical bioenergy production potential of degraded and marginal land was assessed by Wicke, 2011 under different settings and geographical scales. Nijsen et al. [62] indicated that the technical bioenergy production potential of natural and human-induced degraded and marginal land amounts to approximately 90 EJ/yr. A comparison with the results found in the literature indicates that this potential is between the estimate of Dornburg et al. (70 EJ/yr) [15] and the upper end of the range given by Schubert et al. [63] (120 EJ/yr). Accounting for the potential bioenergy production on non-salt-affected arid and semi-arid regions in Sub-Saharan Africa and elsewhere could further increase this potential.

The revegetation of marginal, degraded and saline lands can have substantial benefits, such as increased carbon sequestration, abating erosion, improved water retention and the restoration of ecosystem services. With the (gradual) improvement of soil quality, areas can also provide better livelihoods. In all situations, a balance between those desired positive impacts and the possible negative impacts should be found. Additionally, for example, degraded lands offer grazing grounds, and more vegetation can lead to increased water use. As noted before, optimal solutions need to be tailor-made to regional conditions, but the overall insight is also that, in many cases, such solutions can be identified. The authors of [58,64] highlight that China, with specific attention paid to the loss plateau that was strongly degraded in the past, has large potential for, e.g., switchgrass production on marginal and degraded lands, which is also economically attractive. Wicke et al. focused on the production of biomass on saline lands, illustrated with a range of case studies in South Asia where different planting schemes can lead to the reversal of salinity problems over time and bring soils back into production [65,66]. Similar schemes have been adapted in the Colorado River Delta and in SW Australia where bringing back native Eucalyptus cover reversed the salinity problems. In Africa, the Green Belt movement in the Sahel [67] and the "Just Diggit" campaign in East Africa [68] have similar characteristics. In all those examples, the primary driver for revegetation is abating environmental problems (salinity, erosion, dust bowls, water retention, etc.), but the side-effect is that more carbon is stored in the biomass below and above ground, which allows for increased net harvesting, as well. These schemes represent win–win situations (taking specific regional constraints into account) on large land surfaces on the globe [69,70].

### 3.4. Volumes of Waste Biomass and Displacement

Future availability (as discussed in the previous sections in terms of (future) potentials) will, just as they are today, be related to the production levels of the main commodities that result in waste generation. Overall, organic waste volumes are, over time, increasing due to population growth and economic development. The current uses (especially for food/fodder applications) may reduce due to more stringent quality standards. The currently available estimates from scientific information are unlikely to deliver better projections than the currently available consultancy reports. The key to avoiding the displacement of such flows from higher-value markets is proper monitoring and market data. Price signals are often a good measure to signal that higher-value applications of biomass waste are displaced.

### 3.5. Indirect Land-Use Change, iLUC and Avoiding iLUC and Displacement

All the above-mentioned examples start from the necessity to improve land-use efficiency and agricultural management (both for cropping systems and livestock). Yield-improvement potentials, when comparing the current practices to the best practices, show significant-to-very-large improvement potentials (yield gaps) for many world regions. Such yield gaps are the basis for projected future land use for meeting food demand in many scenario studies (as summarized in Section 3). The speed at which such potentials can be harnessed is at the core of whether biomass production potentials can be developed or not. In the scenarios discussed and the case studies included, yield and productivity increases

over time are the fundamentally important parameters, and progressive assumptions are generally in line with historic developments [22,71]. The following key preconditions are also known: investing in farmer support and training, supporting innovation and investment in agriculture, and monitoring services and research and development. Generally, such investments are profitable when compared to the increased revenues of farming and lower inputs per unit of agricultural output (as well as less environmental damage).

Estimating indirect land-use change due to increased biomass demand for energy has, in the past, been achieved using CGE models, which typically extrapolate future land demand by using demand–supply relations based on historic price developments [72]. This typically results in projecting past land expansion and intensification for future growth in commodity demand, resulting in simulated higher prices and the expansion of land use. Although estimates are always uncertain and are recognized to depend on many assumptions, such approaches have, for years, been unable to capture the impact of dedicated investments and policies to improve agricultural productivity faster than what is historically observed. At the same time, this is the key precondition for iLUC-free biomass production in a selected region without displacing other agricultural outputs (such as food). With the example studies discussed earlier, it can be concluded that meeting multiple sustainability targets is very possible. However, including such parameterization in CGE models has been proven to be challenging because these are typically not disaggregated enough for including such relations. Nevertheless, some recent analyses for Brazil [73] and for Ghana [74] show that when such parameterization is included, the CGE analysis results in (logically) lower prices for modeled outputs and the simulated iLUC effects are avoided. Additionally, in "classic" analyses of iLUC effects, the positive impacts of improving agricultural management can be made visible (few institutions have followed up on this, a good exception being [75]).

The analyses of the yield developments of de Wit for Europe include the carbon stock analyses. The authors of [22,76] illustrate that the further modernization of agriculture in Europe (most notably in Eastern Europe where yield gaps are largest) can be realized in line with historic trends, with more knowledge and means than in the past decades and with subsequent positive effects on the total carbon stored in agricultural soils due to more productive crops for food production and increased land cover of perennials on "released" lands. This is a very obvious example of a win–win situation on a continental scale for future European agriculture and the sustainable use of biobased energy carriers and materials.

## 4. The Demand Side: The Role of Bio-Based Options in Energy and Circular Economy System Scenarios

The extent to which available biomass is actually going to be deployed in future energy systems and as feedstock for renewable material especially depends on its attractiveness versus other options to reduce GHG emissions, and thus, the net GHG avoidance over complete value chains and their respective costs. Given that many energy technologies (such as solar and wind energy, energy efficiency improvements, etc.) also develop and reduce in cost over time (just as biomass options do), the future role of biomass can only be understood from an overall system perspective, which is typically analyzed using integrated energy models and scenarios. Depending on the scenario definitions and assumptions, expected future technology developments, etc., the deployment of biomass in terms of amount and distribution over different uses (heat, power, fuels, feedstock) comes with (sizeable) ranges. In those analyses, the LCA-based data of the value chains, carbon stock impacts (above and below ground) and options to utilize biomass with CCS (BECCS) are to be included to obtain a complete picture of the possible (and optimal) role of biomass in future integral mitigation strategies.

There are plenty of state-of-the-art energy-scenario-modeling efforts at the global, European and country levels that illustrate the possible role of biomass in achieving a low-GHG future. Integrated assessment models on a global scale are comprehensive in showing

the interlinkages between global scenario conditions with respect to land use, technological development and energy-system transition, to name several of the key components. As described in Section 3, sustainable, low-iLUC biomass potentials depend on the scenario conditions chosen. Figure 4, below, provides a synthesis [2] of the projected contribution of biomass to the future world's energy supply in EJ and as a percentage of the total energy supply. Furthermore, the importance of BECCS options and achieving negative emissions is highlighted.

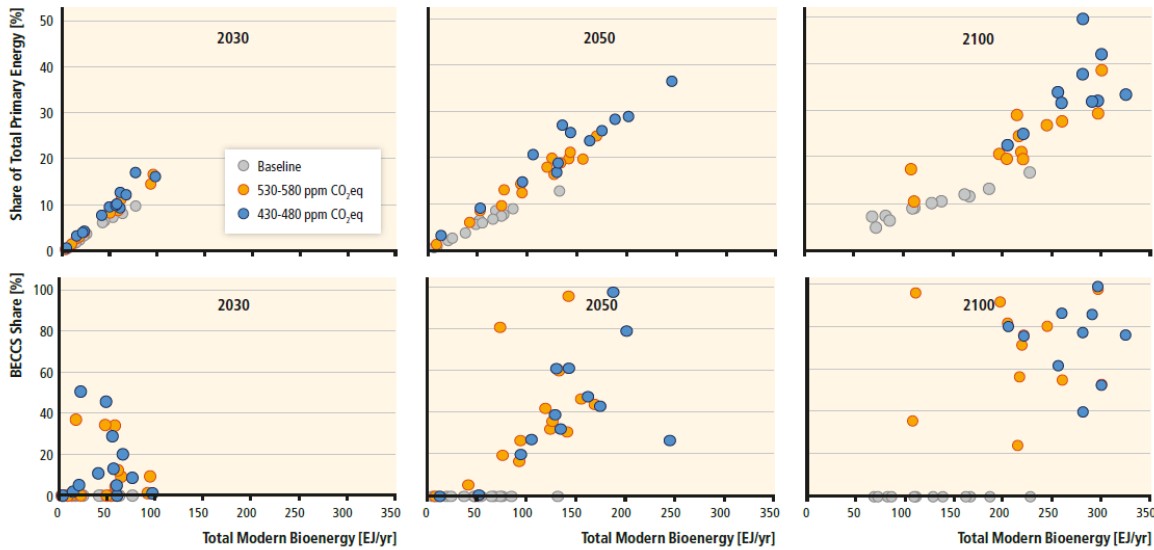

**Figure 4.** Global deployment of biomass for a range of global GHG mitigation scenarios (as assessed by [2]).

Among the reasons why biomass is so important in many mitigation scenarios is its versatility. It can deliver dispatchable power, high temperature, liquid and gaseous (transportation) fuels and renewable feedstock for material (chemicals and construction materials). Furthermore, many biomass conversion pathways are commercially attractive today (e.g., combined heat and power generation from biomass; organic waste conversion to, e.g., biogas; ethanol production from sugar beet, etc.), while biorefinery concepts and further advancements in value chains can make advanced biofuels (e.g., those produced from lignocellulosic material) and platform chemicals in a competitive range in the medium term [30,77,78]. Another key point is that state-of-the-art mitigation scenarios point out that Bioenergy combined with carbon capture and storage (BECCS) can and should deliver negative emissions (carbon taken up during plant growth is released as $CO_2$ during conversion and subsequently captured for geological storage); this is required to meet the 1.5 °C GMT target as agreed in the Paris Agreement. Considering the technological learning that can be achieved in the medium term and the low mitigation costs of various BECCS options (e.g., in industry; see, for example, [79,80]), most scenario analyses result in the deployment of most of the assumed available biomass resources in the future. Negative emissions also allow for some sectors to still emit GHG emissions that would otherwise be (very) costly to mitigate, which also moderates the costs of the mitigation pathways.

### 4.1. The Implications on Implementation and Costs of Mitigation Pathways with and without BBE Options

Country analyses are generally carried out with ESMs with a much higher degree of detail, including specific national data, increasingly with the spatial representation of energy supplies and demand and the inclusion of time patterns. Such recent bottom-up optimization ESMs have been run for countries such as Brazil [81–83], Colombia [84] and the Netherlands [85]; they show consistently that when biomass is deemed available, it will be preferred in optimal-transition scenarios and will meet the GHG mitigation target

over time. The relative convenience of compatibility with existing energy infrastructure (e.g., liquid fuels) and the option to realize negative emissions at relatively low costs have a moderating effect on the overall mitigation cost. Conversely, when the role of biomass is constrained, the targets are generally still met (e.g., with far higher shares of intermittent renewables), but at significantly higher system costs and with more required changes in energy infrastructure and end-use. The latter may, although possible, be an important factor in gaining enough speed in the implementation of mitigation trajectories; rebuilding infrastructure and more fundamental changes in end-use (e.g., vehicle fleets and industrial capacity) has long lead times.

The future shares of biomass used for fuels, feedstock, electricity and heat vary strongly between different scenarios (whether these concern global, European or specific national scenarios) [7]. Most state-of-the-art model calculations and scenario analyses that include well-quantified conversion options from biomass to energy (heat, power and fuels) to a range of biomaterials (most notably base chemicals) result in significant shares of biomass used to cover the future European energy demand when stringent GHG emission targets are to be met [27,28].

In many scenario studies, biomass is, *when available and sustainable* (i.e., produced without indirect land-use change and meeting sustainability criteria), a very attractive mitigation option for industry (feedstock, as well as heat) and the transport sector (aviation, shipping, heavy road transport). This is caused by the (partly current and partly expected) economic performance of a range of advanced biomass conversion routes, such as the production of transport fuels from lignocellulosic biomass and the main platform chemicals (see, e.g., [71] for an overview of value chains, GHG balances and costs).

*4.2. Europe*

Only a few studies at the European scale pay explicit attention to the variability in biomass resource supplies, including biomass materials, BECCS options and carbon stock impacts. Here, we use one of the most recent energy scenario analyses at the EU level by Blanco et al. [27,28], who incorporated all the main biomass utilization options (for biomaterials, only bulk chemicals were considered) and BECCS options. The analysis included the mentioned range of future biomass supplies and incorporated full competition with other GHG mitigation options, such as the large-scale use of solar and wind energy, energy-efficiency improvement, CCS, etc. The model used was the so called JRC TIMES model; it was developed and operated by the Joint Research Centre of the European Commission, and describes the European energy system and all relevant mitigation options (including biomass) in detail and calculates (cost) the optimal scenarios for given assumptions on, for example, the GHG mitigation target, the carbon taxes assumed, the technological performance data, the limitations of the potentials of certain options, etc.

Table 3, below, summarizes the ranges for biomass deployment in 2050 for the different main uses (electricity, transport fuels, heat (both domestic and industrial) and biomaterials). The latter, in particular, is covered by base chemicals such as methanol, ethanol, polyethylene, propylene, etc., because these represent, by far, the largest markets, and thus, GHG mitigation potential. The table reports the primary energy in biomass resources per main market, the overall energy-conversion efficiency per main energy carrier, the final energy delivered (and for biomaterial, an indicative figure for the overall amount in Mton). Table 4 presents the directly related results on total GHG mitigation due to biomass use, negative emissions due to BECCS deployment and carbon stock impact. The total potential contribution of biobased options to the required mitigation efforts of Europe are profound, and may amount to up to half of the total effort when the high values are considered (see Table 4).

**Table 3.** Ranges for biomass deployment for different main uses in Europe in 2050, as derived from scenario analysis with a state-of-the-art energy optimization model [27,28]. The high biomass potential will result in a high impact of biomass use in GHG mitigation and the low biomass potential in low impact.

| Main Product | Primary Biomass Allocated (EJ) | | Net Energy Conversion Efficiency (%) | Final Energy (or Product) (EJ) | | |
| --- | --- | --- | --- | --- | --- | --- |
| | Low | High | | Low | High | |
| Biofuels (second-generation ethanol, DME, Fischer–Tropsch) | 3.15 | 15.6 | 65 | 2.0 | 10 | |
| Electricity (larger scale) | 1.05 | 1.2 | 50 | 0.5 | 0.6 | |
| Heat (larger scale and industrial) | 1.75 | 0 | 90 | 1.6 | 0 | |
| Digestion | 0.7 | 2.4 | 25 | 0.2 | 0.6 | |
| Large scale biorefinery complexes | 0.35 (18 Mton) | 4.8 (246 Mton) | 50 | 0.2 (5.4 Mton) | 2.4 (74 Mton) | (*) |

(*) a crude average conversion factor of 0.3 from primary biomass to chemicals is assumed, based on a range of actual conversion factors for different processes and expert opinions [15].

**Table 4.** Annual emissions (Mton $CO_2$ eq), avoided emissions and net emissions per year per main biomass application for the low-impact and high-impact scenarios in the year 2050.

| Main Product | GHG Emissions, Biomass Value Chains (Mton $CO_2$ eq/yr) | | Avoided Emissions, Fossil Reference Products (Mton $CO_2$ eq/yr) | | Net Avoided Emissions (Mton $CO_2$ eq/yr) (Low Impact Defined as Higher-Emission Biomass Value Chain + Low Deployment; High Impact Defined as Lower-Emission Biomass Value Chains + High Deployment | | |
| --- | --- | --- | --- | --- | --- | --- | --- |
| | Low | High | Low | High | Low | High | |
| Biofuels (second-generation ethanol, DME, Fischer–Tropsch) | 51 | 71 | 205 | 1014 | 154 | 943 | |
| Electricity (larger scale) | 25 | 0 | 84 | 96 | 59 | 96 | |
| Heat (larger scale and industrial) | 24 | 0 | 145 | 0 | 121 | 0 | |
| Biogas | −1 | −42 | 10 | 34 | 10 | 76 | |
| Bulk biochemicals | 3 | 0 | 24 | 332 | 22 | 332 | |
| **Totals** | **102** | **29** | **468** | **1476** | **366** | **1447** | (*) |
| BECCS contribution | | | | | | 678 | |
| Carbon stock increase (average Mton per year up to 2050) (based on [76]. | | | | | 13 | 52 | |
| Total net mitigation impact | | | | | 379 | 2177 | |

(*) Key data used for the calculation of the contribution of BECCS to the total mitigation potential are as follows: carbon content of biomass (dry basis): 50%; heating value of dry clean biomass: 19.5 GJ/ton [15]; total carbon for 7 EJ biomass (low-impact scenario): 179 Mton; total equivalent $CO_2$ (3.67 ton $CO_2$/ton C): 659 Mton $CO_2$; total carbon for 24 EJ biomass (high-impact scenario): 2258 Mton $CO_2$.

## 5. Policies versus Preconditions: An Action Agenda for Positive Action

Bioenergy is currently the largest renewable source globally and in the European Union, and is likely to remain one of the largest RE sources for the first half of this century in most GHG mitigation scenarios. Biomass is of increasing importance for delivering

carbon-neutral feedstock for chemical and other materials, as for liquid and gaseous fuels for key sectors such as aviation, shipping and long-distance truck transport. There is considerable growth potential to make more biomass resources available on a sustainable basis, but it requires active development and policy measures in agriculture, forestry and land use.

- Assessments in the recent literature, as summarized in this paper, show that the resource potential of biomass for energy in the European Union may reach well over 20 EJ/yr (478 Mtoe) by 2050. This includes potential future land availability in Europe, which can materialize when yield gaps are (partly) closed and efficiency improvements in livestock are realized; moreover, 7–52 Mha of arable land and 10–19 Mha of pasture lands could be released in Europe, while meeting future food demand. However, uncertainty exists about important factors such as market and policy conditions that affect this potential. Realizing this potential represents a major challenge but would make a substantial contribution to the EU's primary energy demand in 2050 of one third of total energy supplies.
- State-of-the-art energy and GHG mitigation scenarios suggest that whatever biomass resources are available, they will be used for the various markets mentioned; given the attractive economic performance, biomass use generally lowers overall mitigation costs. An additional driver for biomass deployment is the possibility to deliver negative emissions when the conversion of biomass is combined with carbon capture and storage (BECCS). Many recent scenario analyses point out that such options are necessary to achieve the 1.5–2 °C GMT change target set in the Paris Agreement. The increased technical possibilities provide very good economic prospects for using biomass in future low (zero-to-negative)-emission industries and fuel production with BECCS, and thus, achieving negative emissions. Certain current systems and key future options—including perennial crops, forest products and biomass residues, and waste used in advanced conversion technologies for next-generation biofuels and biochemicals, as well as well-managed first-generation biofuels—can deliver very good GHG mitigation performance, typically with an 80- >90% reduction compared to the fossil-energy baseline. The remaining emissions can be lowered using GHG-neutral energy carriers for agricultural management and agrochemical inputs as a result of the overall decarbonization of the energy system and economy. To achieve such desired impacts and performance, land-use conversion and forest management should enable losses of carbon stocks and indirect land-use change (ILUC) to be avoided, or conversely, carbon stocks to increase over time.
- In order to achieve the high potential deployment levels of biomass for energy without negative displacement effects, increases in competing food and fiber demand must be moderate, land must be properly managed and agricultural and forestry yields must increase substantially, e.g., by improving forest management practices. The expansion of bioenergy in the absence of monitoring and good governance of land use carries the risk of significant conflicts with respect to food supplies, water resources and biodiversity, as well as a risk of low greenhouse gas (GHG) benefits. Conversely, the implementation that follows effective sustainability frameworks could mitigate such conflicts and allow the realization of positive outcomes (for example, in rural development, cleaner and more sustainable agriculture, land amelioration and climate change mitigation, including opportunities to combine adaptation measures). The adaptation to climate change with respect to maintaining and increasing vegetation cover, increasing water-retention functions, avoiding and reversing erosion and salinity and creating more resilient forests and agricultural systems will become increasingly important in the coming decades, when climate change will come with increased and more severe impacts. Increased biomass availability can be an important side-effect of such measures, also improving the economic viability of such (unavoidable) measures.

The key conditions for mobilizing sustainable resource potentials require "smart" agriculture, including precision-farming techniques and integrated concepts with nutrient

recycling (e.g., with the digestion of manure from livestock management), which allow for the combination of higher yields and a reduction in land use per unit of food while lowering GHGs and other emissions and improving soil quality to give it increased carbon storage and better economic performance. The impacts and performance of biomass production and use are region- and site-specific. Therefore, as part of good governance of land use and rural development, bioenergy policies need to consider regional conditions and priorities along with the agricultural (crops and livestock) and forestry sectors.

When biomass is produced via perennial crops planted on lesser-quality land, ecological benefits can especially be achieved. Increased sustainable residue availability from agriculture can also be improved this way. For forest residues, good forest management is the key to mobilizing biomass resources in a sustainable way. Such management schemes result in the maintenance of forest areas and carbon stocks, an increase in forest productivity and the use of forest biomass resources with low carbon-payback times.

*Revisiting the EU Policies with Respect to Biobased Economy*

At the moment, policies at the European level, as well as in several member states, focus on avoiding the potentially harmful impacts of biomass production and use, instead of creating the conditions to increase a sustainable resource base. The current focus is on the available resources (such as waste, and residues from agriculture and forestry) and their (albeit significant) constrained availability.

This can be qualified as tragic, given the importance of biobased options for the mitigation of GHG emissions on a global and European scale and the synergies that can be achieved in doing so, as argued in this paper.

Given the urgency of the mitigation of and adaptation to climate change and the need to resolve multiple sustainability problems (food security, forest preservation, the reversal of desertification, rural development and poverty alleviation and circular materials) simultaneously, it is remarkable that, particularly on a European level, the biomass option is constrained instead of developed sustainably. Aligning policies with respect to renewable energy (and the circular economy), with policies targeting more sustainable agriculture and forestry and GHG-reduction targets overall, can lead to major synergies as well as cost savings. The mentioned size of the contribution that biobased options can make to European mitigation targets alone is in the order of the role of mineral oil today. Note that the EUR 400 billion annual fossil fuel imports of the EU prior to 2020 (and which are increasing dramatically with current fossil fuel price levels) make biomass deployment both highly competitive as well as a major option for the EU to reduce its energy-import dependency and improve energy security. Furthermore, such expenditure can, for a sizeable amount, be shifted to rural regions in Europe, making it a major driver for development and modernization—a flywheel that fits the objectives EU's Green Deal overall.

In order to achieve this, following matters require serious and constructive attention in future EU policy frameworks:

- The importance of strategies and the valuation of the synergies between biomass production and use and other sustainable-development priorities (better agriculture, the management of natural resources, the adaptation to climate change, the circular economy, the affordability of the energy transition and climate change mitigation, and rural development) should be at the core of different combined-policy agendas (agriculture, energy, climate, environment and rural development).
- The importance of having biobased options as a key component of the toolbox to mitigate and adapt to climate change, most notably for industry, the circular economy and transport fuels, as well as providing energy security on a European level.
- Focus should shift from quantifying potential iLUC and displacement risks to mitigation of those risks and enhancing sustainable biomass resource availability. The perspective in standards and rules needs to shift from hedging problems to achieving synergies (governance of land use) and incentivizing practices that prevent or mitigate ILUC.

- The importance of both a good value-chain design and how this fits optimally in a biomass production region in conjunction with other land uses, and achieving win–wins, as discussed, are to be integrated into the RED and the Common Agricultural Policy of the EU. After all, modernization and improving the efficiency and environmental performance of conventional agriculture (and livestock) are essential in themselves, but are also a key preconditions for securing more sustainable biomass.
- No/minimal iLUCs and displacement-risk biomass should be secured via proper monitoring of the overall land use and by enhancing *productivity* in producing regions.
- Flexible biomass feedstock production in relation to fluctuation yields and market demand can become a stabilizing factor in different key markets covering food and biobased commodities, both existing and new.
- Such approaches should be included in certification schemes combined with regional/national monitoring and intervention options. Certification driven by the demand for sustainable biobased commodities sets the pace for conventional agriculture and forestry and provides a lever for improvements. Certain regions and countries can also be excluded the moment required governance is not up to standards.
- The definition of sustainable biomass categories needs to be revised according to the findings presented here. The focus should not be on biomass categories as such, but on the settings in which the biomass sourcing is conducted and how the combined impacts of improved land use, forest management and sourcing turn out on a regional level. With proper sustainability frameworks, synergetic benefits can be the main result. The mitigation of iLUC fits the state-of-the-art sustainability frameworks (covering regions and settings as argued by the FAO and covered, to some extent, by the Roundtable for Sustainable Biofuels [36,86,87]).
- Short term in demonstration schemes should be invested in; "show how" examples are very important in the short term: these can be pilots/demonstrations in selected regions with a size of, e.g., 100,000 hectares that demonstrate how the integrated approaches can be implemented, monitored and scaled-up in different settings.
- The lesson learned from the previous biofuel support schemes, which combined fixed-volume targets with subsidies, is that such policies should be combined with supporting measures to avoid competition for land and other natural resources. Innovation in biomass sourcing interlinked with better management and increased productivity of forest, land and agriculture should be at the heart of such policies. Furthermore, any future targets should be made dependent on the rate of improvement that can be achieved in agricultural and forest management.
- Open biomass and biomass-derived commodity markets and international trade will facilitate developments on European and global scales.
- Schemes such as those mentioned are important and should become fully part of the Green EU Taxonomy.

All of the above is, in addition to being a major agenda for (European) policy, also a priority list for (interdisciplinary) science (agronomy, system modelers, environmental sciences, engineering and social sciences, with respect to mobilizing rural communities and actors.

These key interlinkages between a biobased economy and the optimization of agricultural production (in terms of efficiency and environmental performance), the adaptation to climate change and sustainable land use in general represents a major opportunity for the EU. At the moment, agricultural policy (CAP), renewable energy policy (with a focus on renewables and biobased options) and rural development are aligned. Biobased production schemes can contribute in a major way to GHG mitigation and the displacement of fossil fuels, make the agricultural and forestry sectors more diverse and competitive, and contribute to more sustainable agricultural production overall. Last but not least, nature-based solutions, and thus, biomass production can contribute significantly to adapting to climate change and delivering negative emissions (especially via BECCS schemes).

If achieved, this strategy can deliver new and sustainable economic activity to Europe's rural regions, contribute to a new generation of agro- and forest industries, contribute considerably to energy security, deliver major savings on fossil fuel imports, improve the trade balance of the EU and, last but not least, improve agricultural and forest management practices with the possibility to save considerably on agricultural subsidies.

All in all, a biobased economy is of strategic importance for the EU; biobased options can make a major contribution to lowering GHG emissions, especially for fuels and chemicals; moreover, it can contribute further to negative emissions, and to better land use and agricultural and forest management, in turn, contributing to rural development opportunities. Given the potentials within the EU, biobased options are also a key way forward to enhance future energy security.

**Funding:** This research received no external funding.

**Institutional Review Board Statement:** Not applicable.

**Informed Consent Statement:** Not applicable.

**Conflicts of Interest:** The author declares no conflict of interest.

## Abbreviations

| | |
|---|---|
| BECCS | Bioenergy carbon capture and storage. |
| C | Carbon |
| CCS | Carbon capture and storage |
| CGE | Computable general equilibrium (model) |
| $CO_2$ | Carbon dioxide |
| DME | DiMethyl ether |
| EJ | Exajoule |
| ETC | Energy transition commission |
| EU | European Union |
| GHG | Greenhouse gas |
| GMT | Global mean temperature change |
| Ha | Hectare |
| IEA | International Energy Agency |
| iLUC | Indirect Land-use Change. |
| IRENA | International Renewable Energy Agency |
| IPCC | Intergovernmental Panel on Climate Change |
| LCA | Life-cycle analysis |
| Mha | Million hectare |
| Mtoe | Million-ton oil equivalent |
| Mton | Million tonnes. |
| NGO | Non-governmental organization |
| RED | Renewable energy directive |
| SSP | Shared socio-economic pathway (scenarios) |

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
