# Peer review of "Repairing What Policy Is Missing Out on: A Constructive View on Prospects and Preconditions for Sustainable Biobased Economy Options to Mitigate and Adapt to Climate Change"

_energies, doi:10.3390/en15165955_

Round 1

Reviewer 1 Report

Dear Author,

I thank you for having analyzed this important topic and for having provided the basis, also to the legislator, to be able to help him in the decisions regarding the renewable energy development policies in the coming years. The article is well written and I just ask you to check some typos that follow each other in the text (for example the symbol ° C is often reported as oC).

Best regards,

Nicolò Morselli

Author Response

I thank the reviewer for his kind and positive review; much appreciated and the comments given are exactly why the paper is drafted. 

The manuscript has been revised on various points, including typo's. The revised version is attached with and without track changes.

Reviewer 2 Report

Biomass is the keyword of the bioeconomy and the demand for it is increasing worldwide with the transition to a low-carbon economy. EU must boost its natural resources to increase bioenergy production. I think the topic is interesting and important. However, I would expect deeper and more concise information:

·         The article would need to be situated within a review of studies to date that cover similar areas of research (biomass-based energy self-sufficiency, etc.) so that the approach can take existing knowledge on board and readers can see where it moves knowledge forward. Comments should be situated in the context of existing literature from the previous 3 years. The long term biomass potential can be significant, but depends highly on agricultural intensity. Do the authors see this problem in the context of the Green Deal? What changes in the context of the war in Ukraine? The invasion of Ukraine will complicate the transition path to a net-zero economy.

·         I propose to separate a closer and more distant perspective.  What is the most important at the moment (from the point of view of science, economy, etc.).

·         Too many self-citations.

Hence, I would expect deeper and more concise information. I think the motivations for this study need to be more clearer (compared to the current situation).

Author Response

I thank the reviewer for the comments given. The manuscript has been revised and polished on a range of points. The result (with track changes) is included with this response. 

On the comments given; I respond to throughout the reviewer comments below:

Biomass is the keyword of the bioeconomy and the demand for it is increasing worldwide with the transition to a low-carbon economy. EU must boost its natural resources to increase bioenergy production. I think the topic is interesting and important.

--> Thank you; this is exactly what is tackled by the paper.

However, I would expect deeper and more concise information:

  • The article would need to be situated within a review of studies to date that cover similar areas of research (biomass-based energy self-sufficiency, etc.) so that the approach can take existing knowledge on board and readers can see where it moves knowledge forward. Comments should be situated in the context of existing literature from the previous 3 years. The long term biomass potential can be significant, but depends highly on agricultural intensity. Do the authors see this problem in the context of the Green Deal? What changes in the context of the war in Ukraine? The invasion of Ukraine will complicate the transition path to a net-zero economy.

--> Several responses and changes:

  • The paper does link the current critical attitude towards biomass and bioenergy to what is possible (the scientific basis) and what this should imply for various policy actions (the list of recommendations and synthesis in the last section). The objective has been sharpened and the recommendations edited further to stress this (for example the link to the Green Deal and improving energy security are part of this). The objective of the paper was not to analyze the current energy security situation, but to focus on how the key sustainability concerns around biomass can be addressed. Those sustainability concerns are the basis for the negative attitudes and lack of support (which is stressed in the introduction).
  • The literature basis has been updated further (several older refs removed and several newer ones added, including other institutions). Nevertheless, the key work on future potentials, integrated with analyzing displacement and environmental impacts that focus on how matters can be improved and optimized over time is concentrated in many of the refs included. There is an abundance of literature repeating the potential pro's and con's of biomass production and use, but not providing much more insight in how to develop potentials in a sustainable way. In addition several key review works are important for this paper, referring to 100's of other references (e.g. the IPCC report, the PBL- Strengers report, the Kluts et al RSER paper, etc.). Furthermore, also a review paper (while this is more a position paper) needs to have some constraints on the number of refs that can be included. 

  • I propose to separate a closer and more distant perspective.  What is the most important at the moment (from the point of view of science, economy, etc.).

- See the answers above. The perspective focuses on the sustainability concerns and this is what, in the view of science, is the most important reason why biomass is perceived very critically.

  • Too many self-citations.

- See answers above; the reference basis has been optimized further, but not fundamentally changed because of the focus of the paper and that many key references make the points raised in the best possible way (and quite some of those are not self citations).

Reviewer 3 Report

Despite the interesting content, its structure makes the manuscript not ready for publication at the moment. First of all, but this is my personal opinion, the manuscript is too long. Scientific communication should be concise and to the point, especially literature reviews like this one. Furthermore, many other aspects need to be improved. As follows:

The citation style differs from the format of the journal

English must be improved significantly in order to be published

Line 21: 1.5°C 

Line 96: scenarios

Line 102: algae

Line 123: the list misses references. Where

There are 2 “Section 1”

- Introduction

- Scope of the assessment

Line 162 to 225: the outline of the paper is Clea but it is too long. It should only describe the structure, while the content should be presented in the respective section. It should also name each section. For example:

Section 2 describes the potential of biomass use for energy …

Line 318: why is there a focus only on forest biomass and not on each of the types presented in Table 1? It would make more sense to dedicate a paragraph for each one.

Section 4 is missing

Line 423: this section must be structured better. Why these examples? A table that summarises the examples and the key points of each of them would be useful for readers 

Conclusions are missing

Author Response

I want to thank the reviewer for the comments given! The paper has been reworked and revised on a range of points. The revised version is attached to this response. Specific comments are addressed throughout the review comments copied in below:

Despite the interesting content, its structure makes the manuscript not ready for publication at the moment. First of all, but this is my personal opinion, the manuscript is too long. Scientific communication should be concise and to the point, especially literature reviews like this one.

  • The paper has seen some restructuring following up on more detailed comments below. Nevertheless, the paper is within the length constraints of the journal, so not excessively long. Because many topics need to be addressed in conjunction to discuss the different dimensions of sustainable and effective supply and use of biomass, the paper cannot avoid to tackle them in a way that it can be made clear how biomass can be supplied sustainably in the future and what its effective role can be in the energy transition and industrial transformation. Shorter and more superficial information would reduce the substance of the argumentation, for example on the interlinked improvements of agricultural production, biomass production and avoidance of displacement (which is a core issue), as well as on the changing role of biomass in the energy supply with more and more stringent mitigation targets. These points in combinations are generally not addressed in the policy debate, incentives and strategies (which is a key objective of the paper to tackle that). Given that the paper respects the journals guidelines with respect to length, this results in the set up as presented (although a range of improvements has been implemented; see below).

Furthermore, many other aspects need to be improved. As follows:

The citation style differs from the format of the journal

--> Correct; the required citation style is now implemented.

English must be improved significantly in order to be published

--> The English has been checked once more. Overall writing is most certainly up to standards. 

Line 21: 1.5°C

--> Such errors have of course been corrected. 

Line 96: scenarios

--> corrected

Line 102: algae

--> corrected

Line 123: the list misses references. Where

--> The reference list and citations have been thoroughly revised. 

There are 2 “Section 1”

-->  Correct; this mistake has been corrected.

- Introduction

- Scope of the assessment

--> Has been corrected.

Line 162 to 225: the outline of the paper is Clea but it is too long. It should only describe the structure, while the content should be presented in the respective section. It should also name each section. For example:

Section 2 describes the potential of biomass use for energy …

--> This is a very good suggestion and this has been implemented. More introductory text has been moved to the respective sections.

Line 318: why is there a focus only on forest biomass and not on each of the types presented in Table 1? It would make more sense to dedicate a paragraph for each one.

--> Understand this comment. The positioning of this resource category has been changed and at the same time, next to dedicated cropping systems, this is one of the most complex and most important potential resources, which requires more attention then the, more straightforward, waste streams that are not so much in the center of the sustainability discussions. Forestry however is, which is why different aspects (such as carbon payback) are discussed in more detail.

Section 4 is missing

--> Thank you; section numbering has been revised.

Line 423: this section must be structured better. Why these examples? A table that summarises the examples and the key points of each of them would be useful for readers 

--> Te examples are some of the few available that analyze impacts of biomass production regions in an integrated fashion, combining changes in overall agricultural management with additional biomass production (for different scenario's). Such analyses are scarce and these are some of the clearest illustrating how win win outcomes can be achieved. Other examples could have been included, but that would have made this part lengthier. More examples with less information would loose out on making clear WHY synergies are achieved. This is why the presentation was chosen as presented. The methodological quality of the examples is a key criterion for selecting the examples that in turn underline the diversity in settings that needs to be considered (a point made in the main text).

Conclusions are missing

--> Section 5 is fully written as synthesis and conclusion section, providing answers to the questions and objectives formulated at the start of the paper. It is fine for me to include the word ''Conclusions'' in the section header, but other reviewers have not mentioned this and the current title is chosen because it directly links to answering the questions raised. If this is deemed important; I will include ''conclusion''.

Round 2

Reviewer 3 Report

The author has replied to all comments in a satisfactory way. Can be published in present form.